# Beaded Coronary Aneurysm in Kawasaki Disease

**DOI:** 10.3390/children9101463

**Published:** 2022-09-24

**Authors:** I-Hsin Tai, Kai-Sheng Hsieh, Ho-Chang Kuo

**Affiliations:** 1Department of Pediatric Cardiology, China Medical University Children’s Hospital, China Medical University, Yude Road, North District, Taichung City 40447, Taiwan; 2Department of Medicine, College of Medicine, China Medical University, Yude Road, North District, Taichung City 40447, Taiwan; 3Department of Paediatrics and Kawasaki Disease Center, Kaohsiung Chang Gung Memorial Hospital, Kaohsiung 83301, Taiwan; 4College of Medicine, Chang Gung University, #.259, Wenhua 1st Rd., Guishan Dist., Taoyuan 33302, Taiwan; 5Department of Respiratory Therapy, Kaohsiung Chang Gung Memorial Hospital, Kaohsiung 83301, Taiwan

**Keywords:** Kawasaki disease, beaded coronary aneurysm, CAA, vasculitis, IVIG

## Abstract

Kawasaki disease (KD) is a febrile systemic vasculitis that mainly affects children aged under five years old. The aneurysm formation of the coronary artery is the most common complication after KD. We report a case with multiple coronary aneurysm formation and a special pattern ofbeaded aneurysm after KD and review the form ofcoronary aneurysms in different diseases.

## 1. Case

A 4-month-old infant girlwho presented with intermittent fever without explanation for 14 days was admitted.Laboratorydata revealed leukocytosis (19.5 × 10^9^/L), high sensitivityC-reactive protein (CRP, 1139.04 nmol/L), and thrombocytosis (590 × 10^9^/L). Negative culture reports were found in blood and urine. The empirical antibiotics administration did not achieve clinical improvement. The subcostal view of echocardiography andtransthoracic echocardiography (TTE) revealed three giant fusiform aneurysms of the proximal right coronary artery, along with other smaller aneurysms in the left coronary system. (Figure 1) Acute mitral insufficiency with moderate pericardial effusion was also present. The three beaded aneurysms were subsequently confirmed on a selective right coronary angiogram. (Figure 2) The fever resolved after receivinga high-dose intravenous immunoglobulin (IVIG) infusion. Aspirin with anticoagulant drugs (subcutaneous injection of low molecular weight heparin) administration was initiated and serial echocardiography as well as invasive coronary angiography follow-up were regularly performed sincethe giant aneurysm’s detection. (Figure 3) This report revealeda “beaded aneurysm” as a special medical image for KD.Itsfundamental nature of non-contiguous aneurysms could be a key to identifying antecedent KD vasculopathy compared toother coronary artery aneurysms. Written informed consent was obtained from the participant for the publication of this case report.

## 2. Discussion

Kawasaki disease (KD) is a systemic vasculitis mainly found in children less than 5 years old, particularly in Asian countries;it was first reported by Dr. Kawasaki from Japan in year of 1967. The most common complication is coronary artery involvement and lesions including dilatation, aneurysm formation, and fistula. The diagnostic criteria of KD include fever for more than 5 days and four of the five major clinical presentations including oral mucosa change, bilateral bulbar conjunctivitis, cervical lymphadenopathy, changes in peripheral extremities with desquamation and induration, and skin rash. Nevertheless, the guideline released in 2020 by the Japanese circulation society modified the criteria to ensure that the counting of febrile days is not essential. A high dose of intravenous immunoglobulin (IVIG, 2mg/kg) and aspirin are the standard treatment for KD and was shown to be effective in decreasing aneurysm formation from 20–25% to 3–5% [1,2].The definition of incomplete KD and IVIG-resistance treatment vary to some extent between the country-specific national guidelines. Only IVIG for initial treatment of KD is consistently proposed throughout all guidelines, further therapeutic recommendations vary between the several national recommendations [3].

KD is the leading cause of acquired arteritis in children ≤ 5-year-old in developed countries. The systemically vasculitis-associated mucocutaneous manifestation provides clues to diagnosis due to the lack of specific diagnostic tests. The delayed awareness of KD was likely due to a lack of any principal signs except fever. Although rare, it represents the most prevalent form of atypical KD in young infants. The 2017 Kawasaki disease guideline published by the American Heart Association (AHA) has mentioned that young infants (usually ≤6 months of age) are the most likely to develop prolonged fever without other typical manifestations of KD, making the population more vulnerable to this devastating coronary inflammation and allowing the possibility of eventually developing giant aneurysms because of failure to receive timely treatment of IVIG within 5–9 days of fever.

A pediatrician must check the echocardiography imaging of coronary arteries in infants with prolonged fever and evidence of elevated inflammation markers whenever the reasonable cause is absent because positive echocardiography findings were regarded as a set standard for acute KD accordingly. Coronary aneurysms can occur in up to 25% of KD children without a timely infusion of high-dose IVIG such as in this index case.As coronary aneurysms have become rare due to the widespread use of IVIG therapy for KD, beaded aneurysm is considered to be even rarer. There are no reports on the frequency of beaded aneurysms in KD, but it is presumed to be relatively rare.The final size of coronary aneurysms defines the future risk stratification generally speaking. In small or medium-sized aneurysms (<8mm or Z score <10), spontaneous regression of the aneurysms without significant cardiovascular sequela such as long-term luminal myofibroblastic proliferation is possible. In this case, the subcostal view of echocardiography and TTE revealed three giant fusiform aneurysms of the proximal right coronary artery, along with other smaller aneurysms in the left coronary system (Figure 1a–c). The three beaded aneurysms were confirmed by coronary angiogram is showed in Figure 2. In contrast with smaller aneurysms, giant aneurysms (>8 mm or Z score >10) seldom or never regress over time, which could be observed from the timeline of the index case(Figure 3).

Coronary artery aneurysm (CAA)formation is defined as a localized area of dilatation exceeding the diameter of the adjacent coronary arterial segment by 50%(or 1.5 times greater than adjacent artery).There have been many pathologic diseases including fibromuscular dysplasia (FMD), atherosclerosis, Kawasaki disease, otherautoimmune diseases (such as polyarteritis nodosa, systemic lupus erythematosus (SLE), juvenile idiopathic arthritis (JIA), and scleroderma), trauma (including coronary angioplasty), rheumatic heart disease, coronary artery dissection, syphilis, mycotic coronary emboli, and other conditions that have been implicated in the development of CAA. Atherosclerosis accounts for more than 90% of coronary artery aneurysms in adults, whereas KD is responsible for most cases in children [4].Aneurysmsof KD patients may appear “bead-like” due to multiple coronary segments involved simultaneously, secondary to a prolonged or overwhelming vascular inflammation process.The “string-of-beads” feature was mainly used for the description of the aneurysm in the FMD. The sign is caused by areas of relative stenosis alternating with aneurysms [5].

Yu et al. [6] reported a beaded sample dilatation of coronary arteries in a KD patient with facial nerve palsy. Xing et al. [7] also reported a beaded change inthe coronary artery byusing a 64-slice spiral CT scanner for coronary angiography. The shape of an aneurysm in KD was not well-named or describedin the literature review. CAA is classified into three groups pathologically including inflammatory, noninflammatory and atherosclerotic.Atherosclerosis is the most common cause of CAA in adulthood. In childhood, KD is the most likely cause of CAA (more than 95%). In both conditions of KD and atherosclerosis, the aneurysms are usually multiple site and affect more than one coronary artery segment [8].CAA are often classified into two types as saccular or fusiform according to shape. Atherosclerotic coronary artery aneurysms are usually fusiform, whereas KD-associated and/or post-inflammatory CAA may be of either one type. It is difficult to know whether a CAA was formation congenitally or was acquired as a result of infectious or inflammatory causes due to not regular echocargiography arrangement at birth. The definition of a giant CAA by size in a pediatric patient is generally accepted as larger than 8 mm in diameter (or Z score >10), compared to the definition for an adult giant CAA by size of greater than 20 mm, 40 mm or 50 mm in diameter have all been proposed as a definition in the medical literature, but there is still no clear consensus on how a giant CAA should be defined for adult [9].KD should remain as one of the principal diagnoses in the differential diagnosis of a CAA with rim calcification identified in adulthood because of the prevalence of this disease, particularly in the absence of advanced coronary atherosclerosis or another definite diagnosis including FMD or autoimmune diseases. The use of CT coronary angiography would not be invasive compared to catheter angiography and can be considered as a complimentary imaging modality in children with KD who have aneurysms on transthoracic echocardiography [10].

## 3. Conclusions

We report one KD patient with multiple coronary aneurysm formation and reveal the beaded pattern of saccular/fusiform aneurysm as the special pattern of coronary aneurysm in KD.

## Figures and Tables

**Figure 1 children-09-01463-f001:**
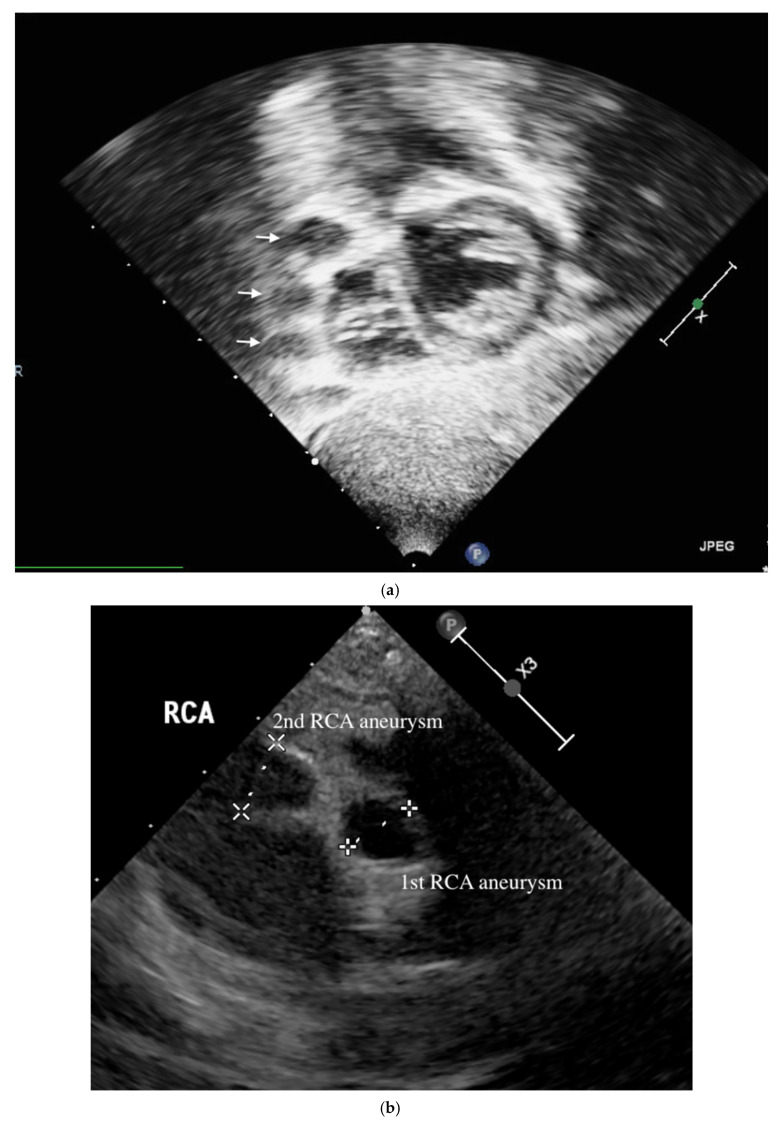
Transthoracic echocardiology on the 16th months after illness of aneurysm. Three RCA aneurysms in a beaded arrangement from the subcostal view of echocardiography (**a**) and transthoracic echocardiogram imaging showed first with second RCA aneurysm (**b**) as well as first with third RCA aneurysm (**c**).

**Figure 2 children-09-01463-f002:**
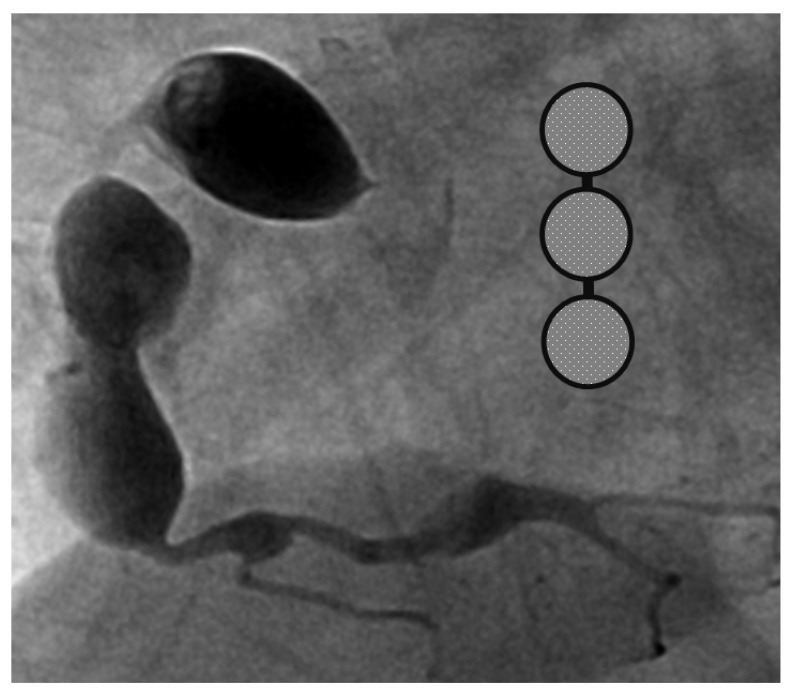
Selective right coronary aneurysm was showed in this figure. Right coronary angiogram showed a beaded pattern of a saccular/fusiform aneurysm.

**Figure 3 children-09-01463-f003:**
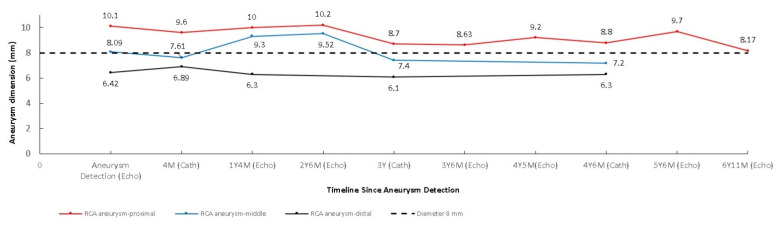
Aneurysm diameter through all clinical course was showed in this figure.The timeline from initial aneurysm detection to long-term follow-up (6 years 11 months). The dashed line indicates a giant aneurysm border (8 mm). Two aneurysms near or exceeding 8 mm in size (red andblue line) and one medium-sized aneurysm (black line). All three aneurysms showed no significant regression whether from echocardiography or invasive angiography sizing.

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
