# Peer review of "Beaded Coronary Aneurysm in Kawasaki Disease"

_children, 2022, doi:10.3390/children9101463_

Round 1
Reviewer 1 Report
I reviewed your manuscript, and cordially make suggestions as followings.
Yellow-colored parts are to be revised according to memo boxes, and red-lined part is to be deleted.
Green-colored parts are to be checked and revised appropriately.
Thank you.

Author Response
We have revised all the comments.
Reviewer 2 Report
1.This case report notes the significance of the early KD diagnostics in young children, in atypical or incomplete form ( echocardiography in children with prolonged fewer!).
2. The images in the article are informative and illustrative.
Author Response
Thanks a lot for the comment.
Reviewer 3 Report
CAA (string-of-beads) in KD is not uncommon in Japan and is not published. If it is to be made into a paper, one should cite the literature on CAA country-specific epidemiological statistics and consider that this case is worthy of reporting.
Incorrect formatting. It is common practice for journals not to cite figures and tables in the Abstract.
1) Rewrite the Abstract as a case presentation.
2) Delete Figures 1-3 from the Abstract, and write the introduction and case presentation separately.
The purpose of the paper is beaded coronary aneurysm formation. The shortcoming is that i t does not state why it is a relatively rare form and how rare it is. At the very least, a literature citation would be necessary. At the very least, a literature search should be conducted to determine how many of the coronary artery dragons formed were multiple. Another shortcoming is that the format of the report is not in the usual case report form at. It clearly deviates from the description format of an academic journal, for example, by citing figure numbers in the abstract. As for the details, there is nothing in particular to point out. The English text is easy to read and there are no problems with the sentence structure. The strong point is the discussion of the differentiation of other diseases. The images a re clear and have value as image presentations. If this is to be a major revision, shouldn't the format of the Children's case report be followed first? If the points raised are corrected, we will review the report again.
Author Response
CAA (string-of-beads) in KD is not uncommon in Japan and is not published. If it is to be made into a paper, one should cite the literature on CAA country-specific epidemiological statistics and consider that this case is worthy of reporting.
--> We have cited the only reference of "country-specific" in Kawasaki disease "
Eur J Pediatr . 2022 Jul;181(7):2563-2573."
Incorrect formatting. It is common practice for journals not to cite figures and tables in the Abstract.
1) Rewrite the Abstract as a case presentation.
--> We have revised the abstract.
2) Delete Figures 1-3 from the Abstract, and write the introduction and case presentation separately.
--> We have deleted figure 1-3 from abstract.
The purpose of the paper is beaded coronary aneurysm formation. The shortcoming is that i t does not state why it is a relatively rare form and how rare it is. At the very least, a literature citation would be necessary. At the very least, a literature search should be conducted to determine how many of the coronary artery dragons formed were multiple.
--> We have added the reference of country-specific of KD.
Another shortcoming is that the format of the report is not in the usual case report format. It clearly deviates from the description format of an academic journal, for example, by citing figure numbers in the abstract. As for the details, there is nothing in particular to point out. The English text is easy to read and there are no problems with the sentence structure.
--> Thanks for your comments.
The strong point is the discussion of the differentiation of other diseases. The images are clear and have value as image presentations. If this is to be a major revision, shouldn't the format of the Children's case report be followed first? If the points raised are corrected, we will review the report again.
--> Thanks for your comments.
Reviewer 4 Report
This case-reporta is clinically interesting
Author Response
Thanks for your comments.
Round 2
Reviewer 3 Report
This article reports a case of multiple aneurysmal beaded aneurysms in Kawasaki disease. The actual frequency is unknown, but it is thought to be relatively rare. If the deficiencies in the article are corrected, this article deserves to be published.
Corrections
1) Incorrect figure numbers: Figure 2 appears first in the text (Case-L10); Figure 1 should be after "in the left coronary system" in Case-L8; Figure 3 should be after "in the left coronary system" in Case-L15. Figure 3 should be placed after "since the giant aneurysms detection" in Case-L15.
2) No mention of the epidemiology of beaded aneurysm: As coronary aneurysms have become rare due to the widespread use of gammaglobulin therapy, beaded aneurysm is considered to be even rarer. Therefore, the sentence "There are no reports on the frequency of beaded aneurysms in KD, but it is presumed to be relatively rare." should be replaced with the following sentence should be added after "such as in this index case" in Discussion-L33.
3) Figure legends are not titled; "Transthoracic echocardiology on the ____ days of illness" should be included in Figure 1, and "selective right coronary aneurysm" in Figure 2. Figure 3 should read "Aneurysm diameter through all clinical course. Figure 3 should include "Aneurysm diameter through all clinical course.
Author Response
1) Incorrect figure numbers: Figure 2 appears first in the text (Case-L10); Figure 1 should be after "in the left coronary system" in Case-L8; Figure 3 should be after "in the left coronary system" in Case-L15. Figure 3 should be placed after "since the giant aneurysms detection" in Case-L15.
--> We have revised it.
2) No mention of the epidemiology of beaded aneurysm: As coronary aneurysms have become rare due to the widespread use of gammaglobulin therapy, beaded aneurysm is considered to be even rarer. Therefore, the sentence "There are no reports on the frequency of beaded aneurysms in KD, but it is presumed to be relatively rare." should be replaced with the following sentence should be added after "such as in this index case" in Discussion-L33.
--> We have revised it.
3) Figure legends are not titled; "Transthoracic echocardiology on the ____ days of illness" should be included in Figure 1, and "selective right coronary aneurysm" in Figure 2. Figure 3 should read "Aneurysm diameter through all clinical course. Figure 3 should include "Aneurysm diameter through all clinical course.
--> We have revised it.